# Development of a Transcriptional Factor PuuR-Based Putrescine-Specific Biosensor in *Corynebacterium glutamicum*

**DOI:** 10.3390/bioengineering10020157

**Published:** 2023-01-24

**Authors:** Nannan Zhao, Jian Wang, Aiqing Jia, Ying Lin, Suiping Zheng

**Affiliations:** 1Guangdong Key Laboratory of Fermentation and Enzyme Engineering, School of Biology and Biological Engineering, South China University of Technology, Guangzhou 510006, China; 2Guangdong Research Center of Industrial Enzyme and Green Manufacturing Technology, School of Biology and Biological Engineering, South China University of Technology, Guangzhou 510006, China; 3Animal Husbandry and Fisheries Research Center of Guangdong Haid Group Co., Ltd., Guangzhou 511400, China

**Keywords:** *Corynebacterium glutamicum*, putrescine, transcriptional factor, biosensor

## Abstract

*Corynebacterium glutamicum* is regarded as an industrially important microbial cell factory and is widely used to produce various value-added chemicals. Because of the importance of *C*. *glutamicum* applications, current research is increasingly focusing on developing *C. glutamicum* synthetic biology platforms. Because of its ability to condense with adipic acid to synthesize the industrial plastic nylon-46, putrescine is an important platform compound of industrial interest. Developing a high-throughput putrescine biosensor can aid in accelerating the design–build–test cycle of cell factories (production strains) to achieve high putrescine-generating strain production in *C. glutamicum*. This study developed a putrescine-specific biosensor (pSenPuuR) in *C*. *glutamicum* using *Escherichia coli*-derived transcriptional factor PuuR. The response characteristics of the biosensor to putrescine were further improved by optimizing the genetic components of pSenPuuR, such as the response promoter, reporter protein, and promoter for controlling PuuR expression. According to the findings of the study, pSenPuuR has the potential to be used to assess putrescine production in *C. glutamicum* and is suitable for high-throughput genetic variant screening.

## 1. Introduction

Putrescine, also known as 1,4-butanediamine, is an industrially important platform chemical that has received a lot of attention for its use as a monomer in the synthesis of the industrial synthetic polyamide plastic nylon-46 (Stanyl^®^, DSM) [1]. Owing to its performance advantages, such as a high melting point, high heat resistance, and superior solvent resistance, nylon-46 is an excellent engineering plastic widely used in textile and automobile manufacturing [2]. Putrescine can also be used as a polymer component and has numerous applications in producing pharmaceuticals, agricultural chemicals, surfactants, and other additives [3]. Putrescine is currently produced primarily through chemical and biological processes. With more emphasis on environmental issues and green and sustainable development, metabolic engineering of microbial cell factories to produce valuable chemicals and fuels is an appealing alternative to chemical syntheses. At present, the main putrescine-producing strains are *Escherichia coli* [4,5] and *Corynebacterium glutamicum* [6,7,8,9,10]. Compared with *E. coli*, *C. glutamicum* lacks the natural ability to synthesize putrescine, necessitating the introduction of heterologous ornithine decarboxylase (ODC) [10]. *C. glutamicum* has advantages in putrescine synthesis, including a strong metabolic flux to glutamate, high putrescine tolerance, and no degradation pathway [3,11].

Traditional microbial-metabolic-engineering methods require extensive optimization and the reconstruction of the metabolic network to create putrescine-producing strains with high yielding ability. The entire process is time-consuming and labor-intensive [5,12,13,14]. Developing high-throughput genetic-engineering tools, such as a biosensor, can help accelerate the design–build–test cycle of production strains (microbial cell factories) and achieve high production of putrescine-generating strains [15,16]. Biosensors are currently the most valuable tool for metabolic engineering [15,16,17,18,19] and are classified into the following four types based on their response and reporting mechanisms: (1) biosensors based on transcription factors (TFs) [20]; (2) biosensors based on riboswitches [21,22]; (3) biosensors based on protein–protein interactions (for example, fluorescence resonance energy transfer systems) [23]; and (4) biosensors based on artificially modified proteins or proteins with specific functions [24,25]. Currently, the two most commonly used biosensors are TF-based biosensors (TFBs) and riboswitches. A TF-based biosensor comprises three parts: a TF; a response element; and a reporter element. The reporter element is typically a fluorescent protein whose expression is controlled by ligand-induced transcription factors (TFs) [26]. In the absence of a proper inducer, the TF binds to the response promoter preventing reporter protein expression. In the presence of a certain number of inducer molecules, TFs bind to them, negating their ability to bind to the response promoter and induce reporter protein expression. Because the fluorescence intensity displayed by the cells is proportional to the number of target molecules produced, high-throughput methods can be used to assess genetic variation or the quality of production conditions. Based on fluorescent genetic circuits, genetically encoded biosensors are powerful high-throughput screening (HTS) tools for obtaining high yielding strains from a library of genetically diverse strains [26,27,28]. TFBs have also been used to successfully monitor changes in target metabolite concentration in real time [17,26], dynamic regulation of pathway enzyme genes [18,29], and adaptive evolution of strains [30].

Metabolite response TFs are primarily involved in metabolite transport and utilization pathways. PsenPut is a diamine biosensor based on the TF CgmR (CGL2612), a TetR-family regulatory protein that regulates the endogenous putrescine transport pathway in *C. glutamicum*. [31]. The developed endogenous TF-based biosensor PsenPut is well adapted and responds to various diamine compounds, but it may interfere with its metabolic network. PuuR, a putrescine-sensing TF, is a negative regulator of the putrescine-utilizing pathway (Puu pathway), which includes enzymes encoded by the *puu* genes in *E*. *coli* K12 that metabolize putrescine to gamma-aminobutyric acid via gamma-glutamylation intermediates [32]. PuuR has a helix-turn-helix DNA-binding motif that binds to the promoter regions of the Puu pathway’s *puuA* and *puuD* genes. PuuR binds to the gene spacer between *puuA* and *puuD* and inhibits *puu* gene expression at low intracellular putrescine concentrations. When putrescine concentration is high, PuuR dissociates from DNA and the *puu* gene is transcribed, thereby initiating the putrescine utilization pathway. Chen et al. recently developed a PuuR putrescine-responsive biosensor in *E. coil* [33]. When the biosensor strain’s growth medium is changed, the whole-cell putrescine-specific biosensor can adjust its response time. Furthermore, the biosensors’ response dynamics and detection range can be altered by changing PuuR expression and manipulating the chromosomal genes involved in putrescine biosynthesis. However, the low dynamic range may limit its application in biosensor systems.

We aimed to create a putrescine-specific biosensor (pSenPuuR) based on the *E. coli*-derived transcriptional factor PuuR as a high-throughput genetic tool for *C. glutamicum* in this study. The *E. coli*-derived TF PuuR has demonstrated the highly specific detection of putrescine among other diamines and amino acids in the diamine synthesis pathway. We further optimized the genetic components of the putrescine biosensor pSenPuuR to improve the signal-to-noise ratio and sensing sensitivity of the biosensor. Finally, we validated the utility of the optimized putrescine biosensor by using different ODCs obtained from the initial phylogenetic tree. The findings indicate that pSenPuuR could be used to monitor putrescine biosynthesis in *C. glutamicum* and could be useful for ODC HTS.

## 2. Materials and Methods

### 2.1. Bacterial Strains and Cultivation Conditions

*E. coli* TOP10 was grown in a Luria–Bertani medium (10 g/L peptone, 5 g/L yeast extract, and 10 g/L NaCl) at 200 rpm and 37 °C for plasmid construction. *C. glutamicum* ATCC 14067 (bought from ATCC) was grown in a BHI liquid medium (37 g/L brain heart infusion (Becton, Dickinson and Co., Franklin Lakes, NJ, USA)) at 220 rpm and 30 °C. The fermentation medium contained 80 g of glucose, 7.5 g of urea, 10 g of soya peptone, 0.5 g of MgSO_4_, 0.7 g of KH_2_PO_4_, and 42 g of MOPS (per liter). For shake flask fermentation of putrescine-producing strains, transformants were selected, picked from the plates, and inoculated individually in 50 mL Erlenmeyer flasks containing 5 mL of the BHI liquid medium, then cultured for 12 h at 30 °C with 220 rpm shaking. The starting optical density (OD) was diluted to 0.2, and the cultures were centrifuged for 2 min at 6000× *g* rpm. The BHI liquid medium was then aspirated, and the cells were resuspended with fermentation medium without glucose and urea. Each culture was then inoculated into 500 mL Erlenmeyer flasks with 50 mL of fermentation medium. The following antibiotics were added at the following concentrations where appropriate: kanamycin at 50 mg/L or tetracycline at 10 mg/L for *E. coli* and kanamycin at 25 mg/L or tetracycline at 5 mg/L for *C. glutamicum*. In addition, 0.5 mM isopropyl-β-D-thiogalactoside (IPTG) was used to induce ODC expression. Appendix A lists all the bacterial strains used in this study, as well as their relevant characteristics.

### 2.2. Construction of Recombinant Plasmids

At first, a two-plasmid sensor system was constructed. The TF PuuR (GenBank: NP_415815.1) and its corresponding response promoters P*puuA*S, P*puuA*F, P*puuD*S, and P*puuD*F were amplified from the *E. coli* genome. PuuR was cloned into PEC-XK99E to obtain pEC-XK99E-PuuR, and the response promoters with the reporter protein mCherry (GenBank: MZ027319.1) were cloned into PEC-T18-mob2. This finally resulted in T18-P*puuA*S-mCherry, T18-P*puuA*F-mCherry, T18-P*puuD*S-mCherry, and T18-P*puuD*F-mCherry, respectively. To construct a single-plasmid sensor system, the PuuR linear expression cassette *Ptrc-PuR-rrnBT1T2* with P*trc* as promoter and rrnBT1T2 as terminator was amplified from pEC-XK99E-PuuR and assembled by homologous recombination into the T18-P*puuD*F-mCherry plasmid in the opposite direction of reporter protein transcription. Following that, we created a series of plasmids to optimize the genetic components of the sensor. The backbone was the single-plasmid sensor pSenPuuR. The original *mCherry* gene downstream of P*puuD*F was replaced with *egfp* (GenBank: AFA52654.1) and *sfgfp* (GenBank: UFQ89826.1), yielding the plasmids pSenPuuREGFP and pSenPuuRsfGFP. To control PuuR expression, the pSenPuuRsfGFP plasmid was used as the backbone, and the original promoter P*trc* was replaced with various native promoters of *C. glutamicum*. The plasmids that obtained were designated as pSenPuuRsfGFP-Psod, pSenPuuRsfGFP-PcspB, pSenPuuRsfGFP-Pddh, and pSenPuuRsfGFP-PaspB. The pEC-XK99E plasmid was used to overexpress ODCs involved in putrescine production. PCR products of all linear vectors, reporter proteins or promoter fragments were first obtained by KOD DNA polymerase (TOYOBO, Osaka, Japan), and then assembled using NEB Builder HiFi DNA Assembly Master Mix (New England BioLabs, Boston, MA, USA). Appendix A list all plasmids and primers used in this study and their relevant characteristics.

### 2.3. Fluorescence Detection

A single colony of the recombinant strain containing the sensor’s associated plasmid was picked from the plate, inoculated into 5 mL of BHI medium (containing 5 mg/L tetracycline and/or 25 mg/L kanamycin as appropriate) and cultured at 30 °C for 12 h while shaking at 220 rpm. The cells were then diluted 1:100 into a 48-well plate containing 900 µL of fresh BHI medium and cultured for 24 h in a microplate thermostatic oscillator at 30 °C while shaking at 800 rpm. To induce PuuR expression for fluorescence detection of recombinant strains with two-plasmid sensors, 0.1 mM IPTG was added. To quantify the dose response of the sensor to exogenous putrescine, the seed culture of Cg14067/pSenPuuR was inoculated into 900 ul of fresh BHI and added with tetracycline and different concentrations of putrescine (0–100 mM). The corresponding antibiotics and 20 mM ligand compound were added according to the experimental needs to evaluate the sensor ligand’s specificity. After the culture was completed, 200 µL of culture medium was taken from each well, and the cells were washed twice with phosphate-buffered saline (PBS; pH 7.4) and resuspended in the same buffer. The fluorescence intensity of the red fluorescent protein mCherry, the green fluorescent protein GFP, and the cell density were measured in the 96-well plates using a BioTek microplate reader and 200 µL of the resuspension solution was used. The wavelengths of excitation and emission for red fluorescent protein and green fluorescent protein were 584/610 nm and 488/520 nm, respectively.

### 2.4. Construction of Putrescine-Producing Strains

The exonuclease-recombinase pairs RecET expression plasmid pEC-XC99E-recET [34] was transformed into wild-type *C. glutamicum* ATCC 14067 (Cg14067). Next, the self-excisable linear expression cassettes dsDNA-P*eftu*-P*argC* and dsDNA-Δ*argF* were constructed according to the RecET-Cre/*lox*P system [34]. To obtain strain P, the linear expression cassette was then transformed into the RecET protein-containing Cg14067/pEC-XC99E-recET receptor strain to complete the replacement of the *argCJBD* promoter on the genome (Appendix A). With PargC-JD-S/PargC-JD-A as the identification primer and wild-type Cg14067 colonies as the control, the positive band size of the promoter that was successfully replaced with P*eftu* was 666 bp. In contrast, the control band size was 541 bp. Then, *argF* knockdown in strain P was completed by the same method to obtain strain PΔ (Appendix A). With ΔargF-JD-S/ΔargF-JD-A as the identification primers and strain P colonies as the control, the control band size was 956 bp, and the positive band size of Δ*argF* was 520 bp. Finally, pEC-XK99E-SpeC_EC_ (Genebank: CP097884.1) was transformed into strains P and PΔ to obtain PUTP and PUTPΔ, respectively. TTTG was used as the PAM sequence in the CRISPR/Cpf1&RecT-based genome fine modification method [35], and the proline codon CCT at position 7 of the *snaA* gene was replaced with TAA to terminate SnaA expression at an early stage. To complete the point mutation of the *snaA* gene, the PΔ/pJYS1Ptac receptor strain was transformed with the crRNA expression plasmid pJYS2_snaA and the 59 bp lagging oligonucleotide O-snaA. To obtain strain PΔD, recombinant *C. glutamicum* containing double plasmids was incubated overnight in an antibiotic-free BHI liquid medium at 34 °C with 220 rpm shaking. Finally, the recombinant strain PUTPΔD was created by transferring pEC-XK99E-SpeC_EC_ into PΔD (Appendix A).

### 2.5. Cell Fragmentation and Putrescine Detection

After the fermentation of the recombinant strain, 500 µL of the fermentation broth was taken from each sample and centrifuged at 12,000× *g* rpm for 3 min at 4 °C to collect the cells and the supernatant separately. Then, 1.5 mL of pre-cooled PBS solution was added, and the supernatant was gently poured off after centrifugation at 12,000× *g* rpm for 3 min at 4 °C. This step was repeated three times to remove the putrescine residue from the supernatant. Magnetic beads (0.5 g) were added to the crush tube before the cells were resuspended in 1.5 mL of PBS solution and lysed with Bead Ruptor 12 (Omni International, Inc., Kennesaw, GA, USA). The samples were centrifuged at 14,000× *g* rpm for 3 min at 4 °C after 10 cycles of lysis. Finally, 1 mL of the cell-free lysate in the supernatant was slowly aspirated for putrescine quantification. The detection of putrescine was performed by precolumn derivatization with danthanoyl chloride, and the derivatized product was analyzed using high-performance liquid chromatography (Waters 2695, Milford, MA, USA) equipped with an ultraviolet detector (Waters 2489, Milford, MA, USA) at 254 nm using a Sunfire C18 (4.6 × 250 mm, 5 mm, Waters, Milford, MA, USA). Acetonitrile and water (gradient elution program: 0 min, 65% acetonitrile; 5 min, 70% acetonitrile; 24 min, 100% acetonitrile; 30 min, 65% acetonitrile) were used as the mobile phase, which was maintained at a flow rate of 1 mL/min and a temperature of 35 °C [31].

### 2.6. Screening of ODC

Currently, phylogenetic tree analysis has become an important strategy for mining new functional enzymes. In order to enrich the ornithine decarboxylase in the putrescine synthesis pathway, we preliminarily screened the ornithine decarboxylase via phylogenetic tree and molecular docking. This method requires a protein sequence with a known function as a probe. Then, the candidate sequences were screened from the huge sequence information by analyzing the affinity of phylogenetic evolution. Finally, the screening range was further narrowed by molecular docking. Li et al., compared seven ODCs from different sources and found that the ODC from *Enterobacter cloacae* (SpeF_ECL_) had the highest specific ODC activity in *C. glutamicum* [9]. Therefore, the amino acid sequence of SpeF_ECL_ was chosen as the probe sequence to mine more ODCs from the National Center for Biotechnology Information (NCBI) database. BlastP results with a candidate ODC sequence revealed similarities ranging from 50% to 90%. One ODC with a similarity of more than 90% was also kept. Some repeated or truncated sequences were removed, leaving 8245 candidate sequences for phylogenetic tree construction. A comparison file spec.fasta was generated by sequence matching of ODCs using MAFFT, and the final spec.tree file was generated by the FastTree construction of a maximum-likelihood phylogenetic tree. The crystal structure of *Lactobacillus* 30a ODC dimer (PDB:1ord) was used as a template by the modeling module function of SWISS-MODEL to obtain the model structure of the candidate sequences, and the pdb file was downloaded. The candidate protein model was docked onto the ornithine molecule with a radius of 12 using the semi-flexible docking software program CDOCKER in Discovery Studio 3.0. The A-chain active sites N153 (N161) and T387 (T395) were chosen, while the B-chain active sites H216 (H224) and Y652 (Y660) were chosen. To assess the reliability of the ligand-protein docking and binding stability, the total energies of the receptor and ligand -CE (-CDOCKER ENERGY) and the receptor–ligand interaction energy -CIE (-CDOCKER INTERACTION ENERGY) were used. The lower the CE value, the less total energy consumed during the docking process and the more stable the docking system. The lower the CIE value, the better the ligand–receptor interaction binding. ODCs with -CE or -CIE significantly higher than that of the probe enzyme were selected from the candidate sequences based on the energy ranking.

## 3. Results

### 3.1. Development of the TF PuuR-Based Putrescine Biosensor

PuuR can bind to the promoter regions of *puuA* and *puuD*, and the gene spacer region between *puuA* and *puuD* contains four DNA-binding motifs; thus, four response promoters were chosen, namely the full-length promoters of *puuA* and *puuD* (P*puuA*F and P*puuD*F) and the truncated promoters of *puuA* and *puuD* (P*puuA*S, P*puuD*S), with mCherry being the reporter protein to test the dynamic range of the sensor (Figure 1a). First, the four reporter plasmids, namely T18-P*puuA*F-mCherry, T18-P*puuA*S-mCherry, T18-P*puuD*F-mCherry, and T18-P*puuD*S-mCherry were transformed into the wild-type strain Cg14067 to obtain Cg14067AF, Cg14067AS, Cg14067DF, and Cg14067DS, respectively, miming the derepressed state of the response promoter in *C. glutamicum* (“ON” state). Based on this, competent cells were prepared, and the regulator plasmid pEC-XK99E-PuuR was transformed into each of the transfected strains (Cg14067AF, Cg14067AS, Cg14067DF, and Cg14067DS) to obtain Cg14067AF/R, Cg14067AS/R, Cg14067DF/R, and Cg14067DS/R, respectively, simulating the repressed state (“OFF” state) of the response promoters in *C. glutamicum*. The fluorescence-to-OD ratios of the recombinant strains were measured after 24 h of incubation at 30 °C under shaking at 800 rpm in 48-well plates. The dynamic range of the promoters was defined as the fluorescence-to-OD ratios of the recombinant strains measured in the derepressed state to that measured in the repressed state (ON/OFF). The fluorescence-to-OD ratios of the response promoters P*puuA*F, P*puuA*S, P*puuD*F, and P*puuD*S in the derepressed state were all higher than those in the repressed state by PuuR, as shown in Figure 1a, with dynamic ranges of 2.18, 1.95, 36.16, and 6.92, respectively. P*puu*DF was thus chosen as the response promoter for the next sensor design.

### 3.2. Dose Response of the Sensor to Extracellular Putrescine Concentration

The checkerboard assay was used to determine the relationship between fluorescence output, expression of the repressor PuuR, and ligand induction of the two-plasmid sensor. In Cg14067DF/R, which contained both the reporter plasmid T18-P*puuD*F-mCherry and the regulator plasmid pEC-XK99E-PuuR, the levels of IPTG and putrescine in the BHI culture medium were independent of each other (Appendix A). mCherry expression was affected by PuuR expression level and exogenous putrescine ligand concentration, as shown in Appendix A. The fluorescence output at 0.001 mM inducer concentration was greater than that at an 0.01 mM and 0.1 mM inducer concentration. When the inducer concentration was 0.01 mM or 0.1 mM and the putrescine concentration was greater than 10 mM, the levels of red fluorescent protein were significantly higher than the background fluorescence.

We built the single-plasmid biosensor system pSenPuuR containing PuuR expression box based on the reporter plasmid T18-P*puuD*F-mCherry because the two-plasmid system is not very convenient for the subsequent engineering biological system and the two-plasmid system is not stable. The constitutive promoter P*trc* regulates PuuR expression, and the transcription direction was opposite to that of the reporter protein. Then, pSenPuuR was transformed into the wild-type strain Cg14067 to obtain Cg14067/pSenPuuR. The dose response of pSenPuuR to putrescine was investigated by varying putrescine concentrations (0–100 mM) in the medium. When there is no putrescine in the culture medium, PuuR binds to the response promoter P*puuD*F, preventing mCherry expression. Putrescine binds to PuuR when a certain amount of putrescine is added to the culture medium. Meanwhile, PuuR was released from the promoter P*puuD*F, allowing mCherry expression (Figure 2a). As shown in Figure 2b, pSenPuuR has a broad response range to the concentration of putrescine in the medium. The fluorescence increased as the putrescine concentration increased, demonstrating a good linear relationship with putrescine concentrations ranging from 0 to 100 mM. The fluorescence increased significantly at a 10 mM putrescine concentration, but it was still not saturated at a 100 mM putrescine concentration. This could be because of the wide range of pSenPuuR responses to putrescine concentrations, or it could be due to the presence of other unknown regulatory mechanisms.

### 3.3. Ligand Specificity of the pSenPuuR Sensor

We tested the pSenPuuR sensor’s response to L-lysine, L-arginine, L-ornithine, 1,3-propylenediamine, 1,4-butylenediamine, 1,5-pentanediamine, and 1,6-hexanediamine to confirm its ligand specificity. The untreated background fluorescence served as the control. Figure 3 shows that pSenPuuR has a significant response to 1,4-butylenediamine but a weak response to 1,3-propylenediamine. However, even at a high concentration of 20 mM, precursor amino acids, such as L-lysine, L-ornithine, L-arginine, and diamines above C4, could not induce mCherry expression above the background level. This finding is consistent with the ligand-specificity test results in *E. coli* [33].

### 3.4. Optimization and Characterization of pSenPuuR

The genetic components of the biosensor must be optimized in order to improve its response characteristics [20,28]. The dynamic range of the biosensor is also affected by different reporter proteins. Therefore, we optimized the reporter proteins in pSenPuuR by replacing mCherry with EGFP and sfGFP, respectively, to generate pSenPuuREGFP and pSenPuuRsfGFP (Figure 4a). The fluorescence-to-OD ratios of different reporter proteins at different putrescine concentrations were measured after the three pSenPuuR plasmids were transformed into Cg14067. As shown in Figure 4b, among the three biosensors containing different reporter proteins, pSenPuuREGFP and pSenPuuRsfGFP fluoresced more than pSenPuuRmCherry at all the tested concentrations of putrescine (1–50 mM). The fluorescence intensity of pSenPuuREGFP and pSenPuuRsfGFP was similar when the putrescine concentration was less than 10 mM, but when the putrescine concentration was greater than 10 mM, the fluorescence intensity of pSenPuuRsfGFP was significantly higher than that of pSenPuuREGFP. The fold change in fluorescence intensity of the three sensors containing different reporter proteins (pSenPuuRmCherry, pSenPuuREGFP, and pSenPuuRsfGFP) at 50 mM putrescine concentration was 8.6, 12.0, and 16.8, respectively. Therefore, sfGFP was chosen as the pSenPuuR reporter protein for further optimization and application.

The checkerboard assay results revealed that altering the expression level of PuuR significantly impacts the final response characteristics of the system. Therefore, we created a plasmid family to regulate PuuR expression levels by varying promoter strength (Figure 4a). PuuR in pSenPuuRsfGFP was expressed at varying levels by the promoters of P*trc*, P*sod*, P*cspB*, P*ddh*, and P*aspB*. The results demonstrate that the fluorescence is still not saturated in the putrescine concentration range of 0–100 mM; thus, the overall dynamic response range of the biosensor cannot be calculated accurately. PuuR expression levels, however, did affect the biosensor system’s response to putrescine. P*cspB* had the highest background fluorescence level (around 1800), and the remaining promoters were not significantly different (all around 1000). However, under 100 mM putrescine concentration, the maximum fluorescence levels were very different, with P*sod* and P*ddh* having the lowest fluorescence levels and P*cspB* and P*trc* having the highest levels (Figure 4c). Given that the ratio of the highest and lowest fluorescence levels corresponding to P*trc*, P*sod*, P*cspB*, P*ddh*, and P*aspB* was 33.7, 3.3, 27.3, 11.3, and 27.5, respectively, pSenPuuRsfGFP-Ptrc was chosen as the final optimized putrescine biosensor for the subsequent screening and verification of ODC.

### 3.5. Construction of Putrescine-Producing Strains

Metabolic engineering of Cg14067 for putrescine production was carried out using CRISPR/Cpf1&RecT and RecET-Cre/*lox*P gene editing technologies. The *argCJBD* operon promoter P*argC* of the L-ornithine synthesis pathway was replaced with a strong constituent promoter P*eftu*, and the ornithine aminotransferase gene (*argF*) of the L-ornithine synthesis pathway of L-arginine was knocked out. The recombinant strain PUTPΔ had a putrescine accumulation of 28.7 mM at the shake flask level after overexpressing ODC from *E. coli* (Appendix A). Therefore, the proline codon CCT at position 7 of spermidine acetyltransferase A SnaA was replaced by TAA, resulting in the premature termination of SnaA translation. The recombinant strain PUTPΔD accumulated 60.0 mM of putrescine, which was 109% higher than the PUTPΔ strain (Appendix A).

### 3.6. Database Mining and Phylogenetic Tree Construction of ODCs

The amino acid sequence of SpeF_ECL_ (Protein No. WP_013097558.1) was used as the probe sequence and was searched using BlastP in the NCBI database. We obtained 8245 candidate ODCs for phylogenetic tree analyses after filtering and screening (Appendix A). A total of 11 ODC sequences were chosen from the 124 ODC sequences in the SpeF_ECL_ branch (Appendix A). Ten ODC sequences were chosen from the other branches of the SpeF_ECL_ branch (Appendix A), and thirty-two ODC sequences were chosen from the remaining branches. From the phylogenetic tree, 53 ODC sequences were chosen for homology modeling and molecular docking. According to the docking results, all candidate proteins were docked successfully with ornithine molecule. Then, five candidate sequences, namely WP_006819050.1 (ODC1), WP_152654430.1 (ODC2), WP_064601238.1(ODC3), WP_016535388.1 (ODC5), and EAP4732819.1 (ODC6), with higher energy and the probe enzyme sequence WP_013097558.1 were chosen for gene syntheses and for subsequent validation (Appendix A). The percentage identity analysis of the five candidate sequences revealed that the identity ranged from 60% to 90% (Appendix A).

### 3.7. pSenPuuR for the Detection of Putrescine Production in C. glutamicum

We introduced the most sensitive plasmid pSenPuuRsfGFP-Ptrc into the previously constructed strain PΔD to demonstrate the application of the PuuR-based putrescine biosensor system in metabolic engineering. The phylogenetic tree screening ODCs ODC1, ODC2, ODC3, ODC4, ODC5, and ODC6 were then cloned into the IPTG-induced pEC-XK99E and transformed into the recombinant strain PΔD/pSenPuuRsfGFP-Ptrc. After 12 h of fermentation, we collected the cells and measured the fluorescence-to-OD ratio of the strains as well as the intracellular putrescine concentration (Figure 5a,b). The level of fluorescence and intracellular putrescine concentration of all strains were compared, and the results showed a strong Spearman’s correlation, with a Spearman’s correlation coefficient r_s_ of 0.67 (*p* = 0.0025; Figure 5c). Therefore, these findings support the use of putrescine biosensors to report putrescine biosynthetic ability.

We also investigated whether the biosensor could be used to report the concentration of putrescine in the fermentation supernatant. We measured the concentration of putrescine in the fermentation supernatant after 12, 24, 48, and 72 h. At each of these four time points, we measured the fluorescence-to-OD ratio of the cells. As shown in Figure 6a,b, for the same strain, the fluorescence intensity gradually increased after 12, 24, 48, and 72 h of fermentation, with an increasing extracellular putrescine concentration. The intensity of the fluorescence was related to the concentration of putrescine. The putrescine yield was comparable between ODC5 and the probe enzyme, which could produce 70 mM putrescine. This finding suggests that the ODC screening strategy based on phylogenetic tree analyses and molecular docking is effective. Finally, we compared the extracellular putrescine concentration and fluorescence ratio among different strains, and the results revealed a strong Spearman’s correlation coefficient r_s_ of 0.63 (*p* < 0.0001; Figure 6c). The results demonstrate that the biosensor could determine the putrescine production titer during shaker fermentation and was suitable for ODC HTS.

## 4. Discussion

Putrescine is a diamine that can be used as a monomer to synthesize nylon-46, an excellent industrial plastic [2,36,37]. Chemical and biological syntheses are the most common methods used to produce putrescine currently available [4,11]. Fermentation syntheses and biotransformation are two of the most common biological methods [37,38,39]. Due to a lack of high-throughput genetic-engineering tools, metabolic engineering strategies’ production efficiency and putrescine yield remain uncompetitive when compared to chemical synthesis methods.

We developed and optimized a putrescine-specific biosensor in *C. glutamicum* based on the *E. coli*-derived TF PuuR. At a high concentration of ligand, the biosensor responded significantly to C4 putrescine, weakly to C3 1,3-propylenediamine, and not at all to diamines above C4 and intermediate amino acids. We systematically optimized the response promoter, reporter protein, and the expression levels of the TF of pSenPuuR to improve the biosensor’s dynamic range and response characteristics. P*puuD*F was the response promoter of the optimal putrescine biosensor pSenPuuRsfGFP, sfGFP was the reporter protein, and P*trc* was the promoter controlling PuuR expression. The results also demonstrate that optimizing the genetic components of the biosensor is critical to improving the response characteristics of the biosensor.

We obtained 53 ODC sequences with different homologies using the SpeF_ECL_ amino acid sequence as a probe, combined with the phylogenetic tree and molecular docking strategy. Finally, we chose six different ODCs to put the optimized putrescine biosensor pSenPuuRsfGFP-Ptrc to the test in metabolic engineering. We looked at the relationship between fluorescence ratio and intracellular putrescine concentration at 12 h and extracellular putrescine concentration at 12, 24, 48, and 72 h. The results revealed a strong Spearman’s correlation between the fluorescence ratio and putrescine production in both vivo and vitro, with correlation coefficients of 0.67 and 0.62, respectively. The putrescine-specific biosensor built in *C. glutamicum* can detect putrescine production during shaker fermentation and is appropriate for ODC HTS.

Currently, various endogenous TF-based metabolite biosensors in *C. glutamicum* have been established, and while endogenous TFs are well adapted, the variety of metabolites that can be detected is limited [17,27]. Therefore, more biological components of metabolite biosensors must be identified, and heterologous biosensors developed and used. Furthermore, naturally occurring TFs are sometimes insufficient for metabolite detection; thus, TFs must be engineered to increase TF sensitivity and change the ligand recognition range [16,40]. TFB engineering strategies currently focus on trial-and-error methods to optimize these genetic components [41,42]. However, because of interaction mechanisms between the biosensor regulatory elements, predicting biosensor performance is extremely difficult. The use of mathematical models to tune biosensor genetic circuits, such as deep learning and machine learning, can provide reasonable solutions while increasing the potential use and impact of biosensor engineering [43].

## Figures and Tables

**Figure 1 bioengineering-10-00157-f001:**
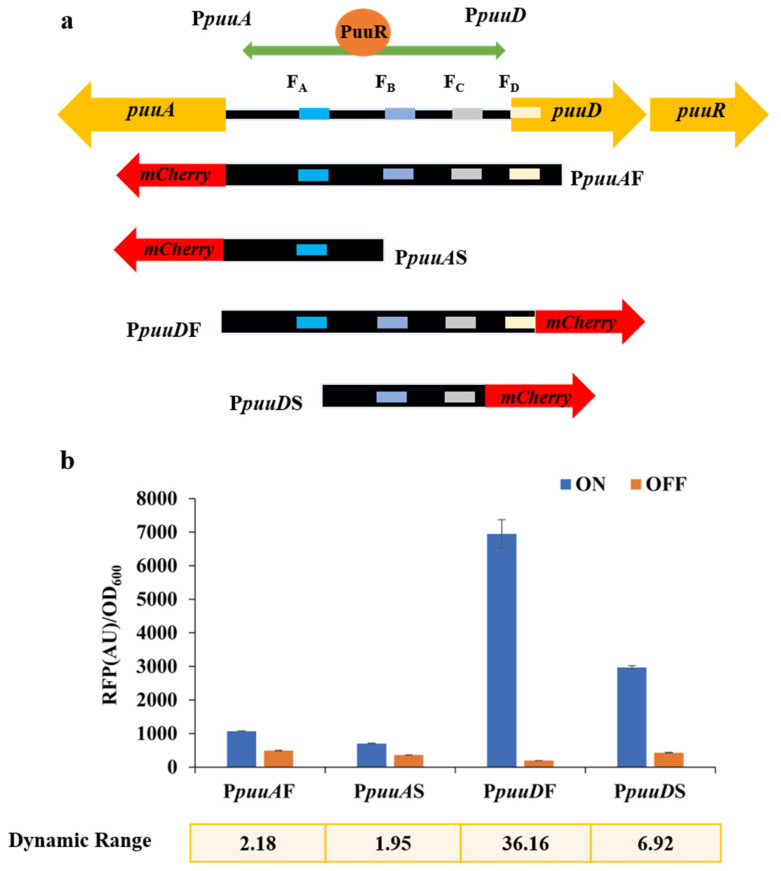
Design of the putrescine biosensor and evaluation of the response promoter. (**a**) Mechanistic model of PuuR regulating *puu* gene expression and selection of the response promoter; (**b**) response promoter evaluation. 0.1 mM IPTG was added to induce the expression of PuuR. Cell fluorescence was normalized by cell density. Data are shown as the mean and standard deviation of independent triplicate.

**Figure 2 bioengineering-10-00157-f002:**
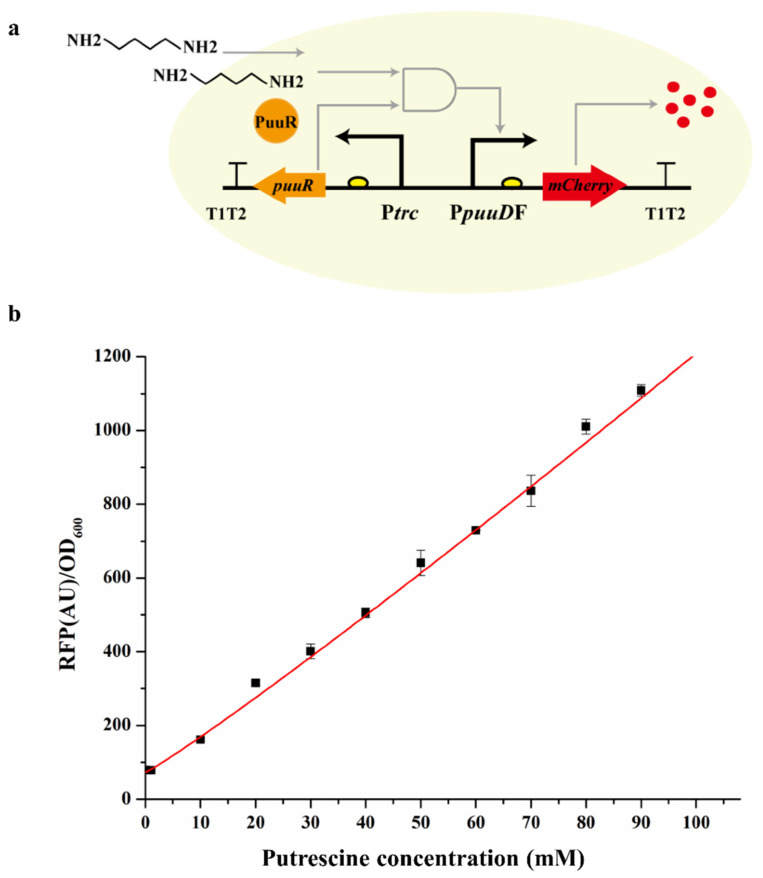
Testing of the single-plasmid biosensor system. (**a**) Schematic diagram of the single-plasmid biosensor system responding to putrescine; (**b**) dose response of the single-plasmid biosensor system to exogenous putrescine. Fluorescence data of PuuR-based biosensors for putrescine concentrations from 0 to 100 mM were fitted to the Hill equation. Values are presented as the mean ± standard deviation of three independent experiments.

**Figure 3 bioengineering-10-00157-f003:**
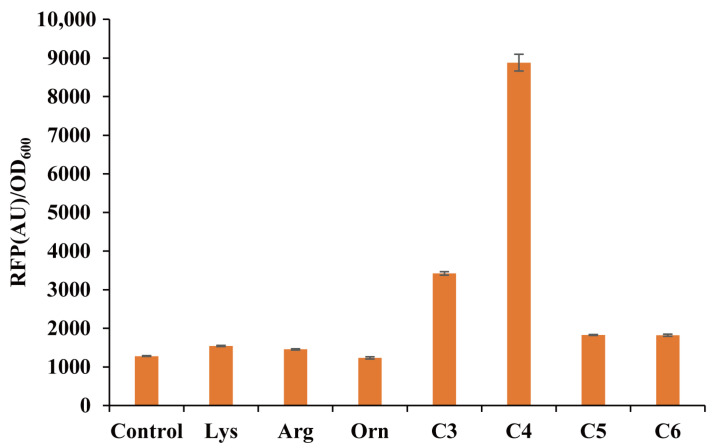
Ligand-specificity testing of pSenPuuR. Amino acids and diamine compounds were detected at a concentration of 20 mM by pSenPuuR. L-lysine (Lys), L-arginine (Arg), L-ornithine (Orn), 1,3-propylenediamine (C3), 1,4-butanediamine (C4), 1,5-pentanediamine (C5), and 1,6-hexanediamine (C6). Values are presented as the mean ± standard deviation of three independent experiments.

**Figure 4 bioengineering-10-00157-f004:**
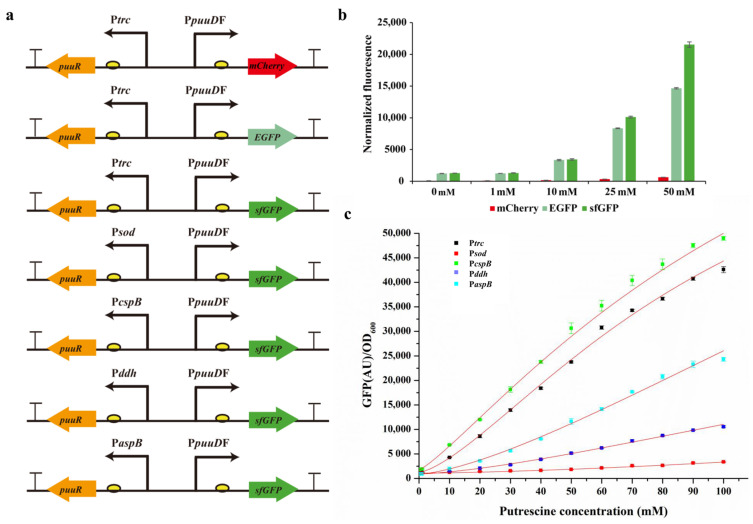
Optimization and characterization of the genetic components of pSenPuuR. (**a**) Constructed plasmid family to optimize the genetic components of pSenPuuR; (**b**) optimization of reporter proteins in pSenPuuR; (**c**) optimization of PuuR expression levels in pSenPuuRsfGFP. Fluorescence data of different expression levels of PuuR-based biosensors for putrescine concentrations ranging from 0 to 100 mM were fitted to the Hill equation. Values are presented the mean ± standard deviation of three independent experiments.

**Figure 5 bioengineering-10-00157-f005:**
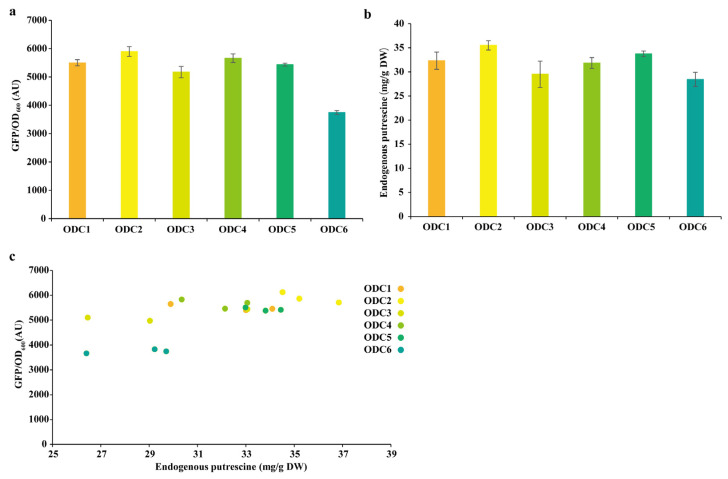
Putrescine biosensor for measuring intracellular putrescine concentration in recombinant strains containing ODCs. (**a**) Fluorescence ratio of the recombinant strain containing ODCs after 12 h of fermentation; (**b**) intracellular putrescine concentration of the recombinant strain containing ODCs after 12 h of fermentation. The intracellular putrescine level was normalized by CDW, which was estimated based on an optical density of 1 corresponding to 0.25 g/L CDW at a 600 nm wavelength. Values are presented as the mean ± standard deviation of three independent experiments. (**c**) Correlation between intracellular putrescine concentration and fluorescence ratio. CDW: cell dry weight; ODC: ornithine decarboxylase.

**Figure 6 bioengineering-10-00157-f006:**
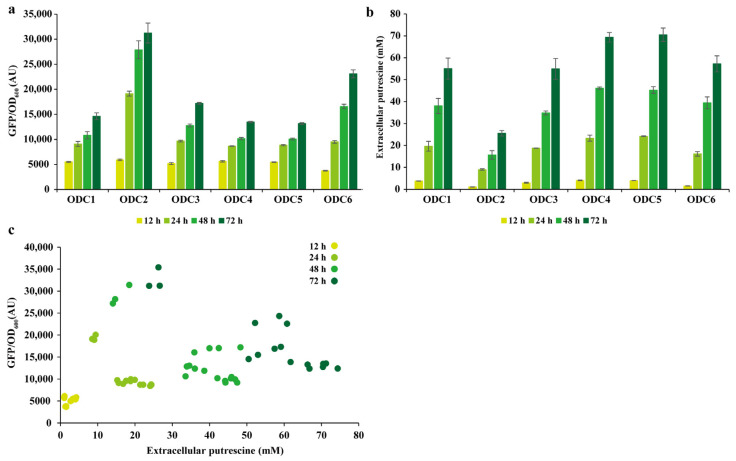
Putrescine biosensor for measuring extracellular putrescine concentration in recombinant strains containing ODCs. (**a**) Fluorescence-to-OD ratio of the recombinant strain containing ODCs measured at different fermentation time points; (**b**) extracellular putrescine concentrations for recombinant strains containing ODCs at different fermentation time points. Values are presented as the mean ± standard deviations of three independent experiments. (**c**) Correlation between extracellular putrescine concentration and fluorescence ratio at different fermentation time points. ODC: ornithine decarboxylase.

## Data Availability

All data generated or analyzed during this study are included in this article (and its Appendix A).

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
