# Peer review of "Development of a Transcriptional Factor PuuR-Based Putrescine-Specific Biosensor in Corynebacterium glutamicum"

_bioengineering, 2023, doi:10.3390/bioengineering10020157_

Round 1

Reviewer 1 Report

Review Report

Development of a transcriptional factor PuuR-based putres-2 cine-specific biosensor in Corynebacterium glutamicum

Nannan Zhao, Jian Wang, Aiqing Jia, Ying Lin, Suiping Zheng

Submitted to be published in Bioengineering.

The authors describe the development of a biosensor in Corynebacterium glutamicum to be used in high-throughput screening to assess and optimise the production of the platform chemical putrescine.

The final construct can indeed be useful to screen for potential genetic variants of Corynebacterium glutamicum to find the optimal putrescine producer. I am not familiar enough with this production in this specific strain to assess if this is a very niche application, or that it can be used for other applications as well. It feels that once developed (here) it can only be applied in 1 specific application, putrescine production in C. glutamicum. Once you found the optimal putrescine producing C. glutamicum, the TFB made itself useless? Maybe I am interpreting that incorrectly, but in that case the authors should make it more clear.

Although I get the phylogenetic tree and molecular docking strategy, I think it could be made a bit more clear/emphasized more why you do that

The language can be improved. Carefully spell check.

The manuscript and the goal is generally clear. I critically read the manuscript and below you can find some comments, questions and suggestions.

Introduction

Clear introduction. I get the difference between production in  E. coli and C. glutamicum.The sentence 46-48 is very vague.

Owing to the complexity of microbial metabolic pathways and the poor functional performance of natural enzymes, the synthetic and catalytic efficiency of putrescine is low.

Poor functional performance of natural enzymes? There is a lot in this sentence but I do not get the message.

78: isn’t developed a better word that established?

79: define CgmR

80: sentence reads a bit weird, consider rephrasing

81: What is meant by endogenous here? PsenPut is developed so not endogenous to the organism.

82: sensitivity increases with carbon chain length. But to a limit I guess? Anything know about that? Range?

91: Is it relevant that they are Korean?

107: …can be suitable for HTS of ODC.

Is that the ultimate goal though? As far as I understand, you need ODC to synthesize putrescine in C. glutamicum. So I get that the system can be used to screen for the optimal ODC to develop the best putrescine producing strain, but once you have developed that, thee ODC screening part is not needed anymore.

Materials and Methods:

112: What’s the source of C. glutamicum?It is mentioned later, better placed here.

117: a triangular flask. An Erlenmyer you mean?

119: ‘controlled’. you mean measured and found as 0.2 or diluted to be 0.2? Unclear what 'control' means.

119: Is it an activated culture (and why?) or are you by this preparing the activated culture?

120: The BHI medium was then aspirated out on an ultra-clean table. The supernanatnt was removed? What does the ‘ultra clean’ part mean and is it relevant?

120-121: the pellet (cells) were resuspended, not the medium.

121-122: inoculation is the introduction of the bacteria to the medium. Not sure if that is meant here. The medium was inoculated with the activated culture, not the other way around.

136: define Ptrc-PuuR-rrnBT1T2

136-137: ‘homologous recombination’: What assembly method is exactly meant here and how was this done? Info is missing.

144: with what technique and how exactly was the promoter replaced?

156: vibrating incubator? A shaker?

Paragraph 2.4 Construction of putrescine-producing strains

Basically this whole paragraph is very unclear.

There are a lot of ‘new’ abbreviations, ‘new’ constructs and terms. Explain why you exactly need those constructs and strains. Maybe an image or so could explain.

191: why first remove 500 µl and then centrifuge? Isn’t the whole reason to centrifuge not separating pellet and supernatant? Feels a bit redundant.

195: ‘to remove putrescine from the supernatant’, does that mean you wash the cells to remove extracellular putrescine?

201: ok to refer to previous reports but maybe describe briefly.

203: Enterobacter cloacae in italic

223-224: ‘Based on the energy ranking, some genes were selected for gene synthesis and for subsequent validation’, vague sentence. Be more precise.

Results

223: transferred -> transformed?

257: Need to briefly explain checkerboard assay? Or is it common enough to assume everybody knows what it is?

320: intensity fold … was …. Is that correct English?

347: ‘pSenPuuRsfGFPPtrc’, was this done before parts 3.2 and 3.3? or is it coincidence that the most optimal trc was used initially? If promoter strength was done first, then maybe put that forward in the text?

Paragraph 3.6 Database mining and phylogenetic tree construction of ODCs

You mention 6 candidates in the next paragraph?

384: ρ (rho) or rs?

Figure 5: use different colours.

Figure 6: colours do not correspond

Figure 6a: sfGFP?

Discussion

432: what exactly is meant by evolutionary tool?

446-447: isn’t the intracellular and extracellular somehow correlated? Is that known or measured? Seems relevant.

450: in vitro?

References

Reference 6 and 7 are the same

Supplemental information

Table S1: bought or purchased from, not buy. Or just mention supplier.

Table S2: I see sequences, not characteristics of the primers

Reviewer 2 Report

In this manuscript, the authors developed and optimized a putrescine-specific fluorescent biosensor based on the E. coli–derived transcription factor PuuR for use as a high-throughput genetic tool for C. glutamicum and  improving its specificity and for putrescine biosynthesis. These studies allow to increase the efficiency of biological production of putrescine.

The study are well designed and all necessary experiments have been performed. The introduction contains adequate information about the study.

1) However, the entire manuscript is very difficult to read. It contains many stylistic errors and convoluted sentences, making it difficult for the reader to understand what the authors mean. Here are the examples:

- line 120-121 – “The BHI medium was then aspirated out on an ultra-clean table and resuspended in a glucose- and urea-free medium”. – In both of these sentences, BHI medium is the subject, implying that the BHI medium was first sucked off and then dissolved…

- line 430-432 - In this study, we developed and optimized a putrescine-specific biosensor based on the E. coli–derived TF PuuR in C. glutamicum, with the aim of improving the specificity of a HTS and evolutionary tool for putrescine biosynthesis, thereby increasing the biological yield of putrescine.

2) In addition, the authors are very imprecise in their descriptions, for example:

- line 46-48 – “Owing to the complexity of microbial metabolic pathways and the poor functional performance of natural enzymes, the synthetic and catalytic efficiency of putrescine is low.” – the question is: where (or perhaps when) is this situation. It is not written in such a way that it results from the previous sentence. The sentences have no logical connection/continuation.

- similar situation in lines 104-105

- similar situation in li lines 368-369

- line 433-434 – what is the ligand molecule? The reader has to guess what the author meant, because such a term appears for the first time in the text, although these are Conclusions  

3) In Materials and Methods, section 2.5 Cell fragmentation and putrescine detection:

- line 200-201 – “Precolumn derivatization methods and detection conditions have been reported previously” – this method should be briefly described, the reference is not sufficient.

4) SpeF and SpeFECL - the meaning of the abbreviations is nowhere explained in the manuscript.

5) The descriptions of the figures should be more precise. This will make the article easier to understand for the reade, e.g.:

- In the description of Fig1 it should be described what the inducer was and at what concentration it was added, although this is stated in Mat and Met.

6) The experiment showed in Fig2 is not described in Materials and Methods. Overall, the description in results (section 3.2 Dose response of the sensor to extracellular putrescine concentration) is also unclear.

7) The experiment showed in Fig 4a should be better explained (lines 333-337)

8) The legend in Fig. 4b is incorrect - the colors do not match what is shown in the diagram. The same is true for Figs6a and 6b.

9) FigS1b (supplementary) - it should be added in the description that this is an experiment with exogenous putrescine.

 In conclusion, the manuscript should be carefully read and revised, paying special attention to stylistic, grammatical, and logical errors, not only those mentioned by the reviewer in the examples. The descriptions of the experiments should also be more comprehensible.

Round 2

Reviewer 2 Report

The authors modified the manuscript according to the suggestions.

They have used a paid editing service to modify the manuscript in depth, and  improved some technical wording and unsatisfactory descriptions to make the experimental results more convincing.